# Potential prognostic factors for hamstring muscle injury in elite male soccer players: A prospective study

Ismet Shalaj[1⊙], Masar Gjaka[2,3⊙], Norbert Bachl[1⊙], Barbara Wessner[1,4⊙], Harald Tschan[1⊙*], Faton Tishukaj[1,5⊙]

**1** Centre for Sport Science and University Sports, University of Vienna, Vienna, Austria, **2** Department of Human Movement and Sport Sciences, University of Rome "Foro Italico", Rome, Italy, **3** Department of Sport and Movement Science, University for Business and Technology, Pristina, Kosovo, **4** Research Platform Active Ageing, University of Vienna, Vienna, Austria, **5** University of Pristina "Hasan Prishtina", Faculty of Physical Education and Sports, Pristina, Kosovo

⊙ These authors contributed equally to this work.
* harald.tschan@univie.ac.at

**Data Availability Statement:** The anonymized data set has been uploaded as Supporting Information.

## Abstract

Hamstring injuries remain the most common injury type across many professional sports. Despite a variety of intervention strategies, its incidence in soccer players playing in the UEFA Champions League has increased by 4% per year over the last decade. Test batteries trying to identify potential risk factors have produced inconclusive results. The purpose of the current study was to prospectively record hamstring injuries, to investigate the incidence and characteristics of the injuries, and to identify possible risk factors in elite male soccer players, playing in the Kosovo national premier league. A total of 143 soccer players from 11 teams in Kosovo were recruited. To identify possible prevalent musculoskeletal or medical conditions a widespread health and fitness assessment was performed including isokinetic strength testing, Nordic hamstring strength test, functional tests, and a comprehensive anamnesis surveying previous hamstring injuries. On average 27.9% of the players sustained at least one hamstring injury with three players suffering bilateral strains with the re-injury rate being 23%. Injured players were significantly older and heavier and had a higher body mass index compared to non-injured ones (p < 0.05). There was a lower passing rate in the Nordic hamstring strength test and a higher injury incidence among the previously injured players compared to non-injured ones (p < 0.05). Except for hamstring/quadriceps ratio and relative torque at 60˚/sec (p < 0.05) for dominant and non-dominant leg, there were no other significant differences in isokinetic strength regardless of the angular velocity. No differences were observed for functional tests between cohorts. Regression analysis revealed that age, Nordic hamstring strength test, previous injury history, and isokinetic concentric torque at 240˚/sec could determine hamstring injuries by 25.9%, with no other significant predicting risk factors. The battery of laboratory and field-based tests performed during preseason to determine performance related skills showed limited diagnostic conclusiveness, making it difficult to detect players at risk for future hamstring injuries.

**Funding:** Open access funding provided by University of Vienna. No additional external funding was received for this study.

**Competing interests:** The authors have declared that no competing interests exist.

## Introduction

It is an international standard in elite soccer for players to undergo preseason medical examinations and fitness testing. The major aims of these health and performance screening measures are on the one hand to detect and prevent disease. On the other hand, a profile of strengths and weaknesses of motor abilities and skills across a variety of fitness and performance components of individual players is generated to provide baseline physical measurements for designing and implementing training programs, attempting to increase performance and potentially decrease the risk of injury. In fact, a vast number of studies, dealing with the overall injury incidence in soccer players have reported a high number of injuries, particularly for the lower limbs, regardless of the competing level [1, 2]. In this regard, country-specific differences and potential injury mediators have been reported [3]. The question however is, if the same set of laboratory and field-based tests performed to determine performance-related skills can be used to prospectively detect players at risk for future hamstring injuries and furthermore enable the selection of appropriate individualized injury prevention strategies to reduce the perceived risk [4–7].

Hamstring strain injuries (HSIs) are extremely common in soccer players resulting in essential losses in training and playing time [1, 8], impairing individual as well as team success and inducing huge financial costs for soccer clubs and federations [9–11]. Based on prospective studies performed in elite male soccer players in Europe, HSIs accounted for 12–16% of all injuries sustained [12, 13] and for 37% - 47% of all muscle injuries [1, 14], representing the most common non-contact injury. It has been calculated that approximately one in five players will suffer a hamstring injury in any given season. On average, for a professional player, 18 days and three matches are missed due to HSIs, whereas on a club level 90 days and 15 matches will be missed per season, corresponding to 25% of players' absence in games [15]. A lower injury rate of HSIs has been reported among lower division soccer clubs [16]. Other studies show an exceptionally high recurrence rate of HSIs affecting approximately 30% of players [17]. However, the recurrence proportion differs based on the playing level of soccer players, with lower-level players suffering higher injury recurrence proportions [16]. It has to be noted that over the last decade, a considerable number of strategies to prevent HSIs have been designed [18–21]. Although there is distinct supportive evidence that specifically strategies which include eccentric hamstring exercises can prevent HSIs in elite soccer players [22], the incidence of HSIs in soccer players playing in the UEFA Champions League has increased by 4% per year over the last decade [23]. Based on recent research, 57–72% of all HSIs occurred during high-speed running activities with the long head of the biceps femoris, which is the muscle positioned laterally within the hamstring complex, being the most common injury site [24]. However, other high-speed and power activities such as acceleration, deceleration, kicking, and change of direction activities can also be a trigger for HSIs [25–27]. Although the mechanisms for HSIs are not fully understood, the majority of studies and modelling techniques suggest that the most likely timing of the injury is the late swing-phase of high-speed running. It has been shown that large passive torques affect the hamstring muscles during both the initial and the late swing phase and the mechanisms of injury may relate to the timing of these peak torques [24, 28–30].

A considerable number of researchers have analyzed modifiable and non-modifiable risk factors providing various conclusions. With respect to non-modifiable risk factors, previous HSIs and increasing age have shown to heighten the risk of suffering a HSI by 11.6, and 1.4 times, respectively [31]. Concerning modifiable risk factors, the number of factors potentially causing HSIs is decisively greater, including lower limb eccentric muscle strength, hamstring/quadriceps (H/Q) ratio, proprioception, muscle imbalances, flexibility, balance and agility

[32–36]. With the exception of previously suffered HSIs [14, 37], meta-analytical and literature review research have reported inconclusive results for the aforementioned risk factors [14, 23, 38–40].

In light of the high incidence, high rates of recurrence, and financial burden, the identification of risk factors and injury mechanisms related to HSIs is essential to detect injury-prone players and to design measures to successfully prevent injuries. Such individualized conditioning programs must be based on a thorough analysis including assessments to identify areas of weakness within a player's performance profile. Therefore, this study aimed to prospectively record HSIs in the Kosovo national premier soccer league to investigate the incidence and characteristics of HSIs in male elite soccer players during an entire soccer season. Additionally, this study aimed to identify possible risk factors for HSIs in elite Kosovar soccer players in an adequately powered prospective cohort study. For the present study, it was hypothesized that players with one or more risk factors would be more prone to suffer from HSIs.

## Materials and methods

### Design & participants

This observational prospective cohort study on physical and injury-related variables was approved by the Ethics Committee of the Medical Faculty of the University of Pristina (ref number 4738) and was completed during the 2013/2014 season of the Kosovo national premier soccer league. It is part of a long-term study consisting of different modules. A previous publication focused on the overall incidence, severity, and injury type in the highest competing level in Kosovo [3]. The same subjects participated in the current study focusing on hamstring injuries which represent the second most common injury subtype following knee injuries. In total 143 elite male soccer players from 11 teams of the Kosovo national premier soccer league fulfilled the inclusion criteria, provided written informed consent to participate, and completed all preseason anthropometric and functional performance test measures. Players suffering from acute lower limb injuries or recovering from recent surgical interventions (within the last 12 months) were excluded. To identify possible prevalent musculoskeletal or medical conditions as well as performance impairments that could possibly increase the risk of HSIs, a widespread health and fitness assessment was performed. As part of this assessment, a comprehensive anamnesis was carried out asking players to describe the nature and date of any previous hamstring strains using the Injury Report Form provided for by the Oslo Sports Trauma Research Centre [2] and to list any residual problems (number of previous hamstring strains; time since most recent injury; training and match time lost through injury). During the prospective study, all participants playing in the entire autumn and spring season were observed by the clubs' medical staff (either physiotherapists or medical doctors) previously familiarized with the proper use of the questionnaires. To identify total body injury occurrence from where data collection concerning hamstring injuries were extracted, the questionnaire compiled by the Injury Consensus Group established by the FIFA Medical Assessment and Research Centre was used [41]. Continuous reporting, help, and feedback were provided by the medical staff on a weekly basis.

### Anthropometric assessment/body composition

Prior to the physical fitness testing, anthropometric measurements were carried out according to international standards for anthropometric assessment [42] including measurements of body mass to the nearest 0.1 kg and body height to the nearest 0.5 cm, using a portable scale and stadiometer equipment (Seca, Hamburg, Germany). Thereafter, the body mass index was calculated [(BMI = weight (kg) / height (m$^2$)].

## Fitness assessment

All players underwent a comprehensive fitness test battery, which aimed to identify performance impairments that could be associated with an increased risk of suffering a HSI. Subjects were instructed to avoid any strenuous physical activity or training on the two days prior to testing to minimize the possible effect of fatigue. Prior to all the physical performance tests, except for the sit-and-reach test (SRT), a 15-minute warm-up protocol consisting of various running drills without the ball was conducted. Warm-up was not performed before the SRT to avoid any warm-up effect. This test battery is composed of a set of valid, reliable, and sensitive fitness tests, measuring performance characteristics and/or potentially modifiable risk factors, which might have an influence on HSIs. The same assessor, either physiotherapists or sports scientists trained in measurement techniques and protocols, assessed all players. All tests were completed within one week, beginning with the isokinetic torque measurements performed on one day, the SRT, and the Nordic hamstring strength test (NHST) carried out in the listed order on another day, and finally, the countermovement jump, speed, and agility tests performed on the last testing day. All tests were performed before season kick-off.

## Muscle strength and power

To detect imbalances resulting in lowered H/Q ratios, all players were tested using the Biodex System 3 isokinetic dynamometry (Biodex Medical Systems, Shirley, NY, USA). Drawing on the recommendations by Croisier et al. (2008) [43], the testing protocol included several maximum concentric and eccentric exertions of both the hamstrings and quadriceps muscle groups over a range of motion of 100 degrees, interspersed by 2 minutes of passive recovery. Detailed information concerning the test protocol with the repetition number for each velocity is listed in Table 1.

In addition, all participants were tested making use of the simple NHST [44]. For this test, the players are on their knees with their ankles fixated on the floor. The participants are instructed to lower their upper body toward the floor in a slow and controlled manner, keeping their backs and hips straight. Hamstring strength was classified as either failed or passed depending on whether the subjects could hold the position beyond 30 degrees from the vertical starting position.

Countermovement jump performance was additionally assessed as a representative measure of leg muscle power. Starting from an upright standing position with their feet shoulder-width apart and hands on their hips throughout the jumps, participants squat down to a knee angle of approximately 90 degrees before jumping up vertically as explosively as possible. Jump heights and indices of movement efficiency and symmetry were calculated automatically using the Leonardo Mechanograph® ground reaction force plate (Leonardo Mechanograph, Galileo Novotec Medical GmbH, Germany). The subjects performed the test three times, separated by 2 minutes between each trial, and the best result was used for further analysis [45].

**Table 1. Crosier's isokinetic test protocol.**

| Muscle group | Contraction mode | Angular velocity (°/s) | Repetitions |
|---|---|---|---|
| H / Q | Concentric | 60 | 3 |
| H / Q | Concentric | 240 | 5 |
| H | Eccentric | 30 | 3 |
| H | Eccentric | 120 | 4 |

H (Hamstrings); Q (Quadriceps).

## Speed and agility

Performing change-of-direction maneuvers explosively and efficiently as quickly as possible are fundamental performance aspects in soccer, with these activities being associated with an increase in hamstrings injury susceptibility [18]. In the current study, the Illinois Agility Test (IAT), which has shown to be a reliable and valid test in team sports [46], was used to evaluate change-of-direction speed. This course is 10 meters long and 5 meters wide with four cones in the middle which are placed at a distance of 3.3 meters from the starting line to the end line. Starting from a laying position, the participants were instructed to accomplish the course as quickly as possible. The players had to sprint 10 meters, return to the starting line, make another turn and weave in and out of the 4 markers towards the end line as well as towards the base line, and then complete two 10 meter sprints to finish the agility course. Time was recorded using an electronic timing system (Brower Timing, USA) with infrared timing gates positioned at the start and the finish line. The best time of 3 trials was used for statistical analysis.

In addition to the IAT, a 20- and 40-meter sprint test was used to determine the players' acceleration and speed. The tests were performed on a natural soccer field. Players started in a standing position, with the toes of the front foot placed on the starting line which was 50 cm behind the first photocell gates. The players were instructed to start at will but were not allowed to perform fluctuation movements before starting. The participants were timed using a wireless timing system (Brower Timing Systems, USA). The best result of three trials separated by 5 minutes of rest was used for analysis.

## Flexibility

The SRT was chosen to assess flexibility, as it is a frequently used, valid, and sufficiently reliable field test to measure the flexibility of the lower extremities [47]. To perform the SRT, the athletes sat on the floor with their legs fully extended and the soles of their bare feet resting against a purpose-made sit-and-reach box. The athletes placed one hand on top of the other and were instructed to slowly lean forward as far as possible along the measuring line. The distance reached by the athletes' fingertips (cm) was recorded. To eliminate warm-up effects, the SRT was only performed once.

## Statistical analysis

Descriptive statistics, such as age, anthropometric data, and physical performance tests were used for baseline characteristics and means ± SD were calculated. Players with a previous hamstring injury as well as playing position are presented in percentages. Differences between groups are calculated by independent samples t-test. Comparisons in frequency distribution for non-parametric variables between groups as determined by the $Chi^2$ test are presented in percentages and p-values. Relative isokinetic values are calculated by dividing absolute torque measurements by the body mass of the players. In order to explain the prediction contribution to the overall hamstring injury rate as the dependent variable, multiple linear regression models were developed. The effect size, acknowledged as the quantitative measure of the magnitude of an observed occurrence, was calculated and interpreted as follows: small (0.2–0.3), medium (0.5) or large (>0.8) [48]. The confidence intervals for training, match, and total exposure time with respect to the injury incidence were analyzed using the normal approximation model described by Sahai (1993) [49], whereas for the remaining parameters 95% confidence intervals (CIs) were applied. The exact *P* values are reported throughout the manuscript and values ≤0.05 were acknowledged as statistically significant. Data were analyzed using the

Statistical Package for Social Sciences (SPSS version 21) and the effect size was calculated using G*Power (version 3.1).

## Results

### Player baseline characteristics

The total sample of 143 players was composed of 7 goalkeepers (4.9%), 27 internal (18.9%) and 20 external (14.0%) defenders, 18 central (12.6%) and 23 external (16.1%) midfielders, 20 wingers (14.0%) and 28 strikers (19.6%). The players' age, body mass, height, and BMI were 23.2 ± 4.1 yrs, 74.2 ± 6.7 kg, 180.0 ± 5.3 cm, and 22.9 ± 1.7 kg·m$^{-2}$, respectively. 129 (90.2%) players specified their right leg, and 14 (9.8%) their left leg as the dominant leg, which was defined as the leg they preferably used to kick the ball.

### Training and match exposure

Over the entire study period, a total of 36,833 hours of exposure time, consisting of 31,998 hours of training and 4,834 hours of match play, were registered. On average, players participated in 25.3 ± 4.0 matches and attended 149.2 ± 14.3 training sessions. This resulted in a mean exposure time of 257.6 ± 24.9 hours, including 33.8 ± 8.9 hours (13.1%) of match play and 223.8 ± 21.5 hours (86.9%) of training per player.

### Hamstring injury incidence

There were 43 HSIs in total (16 training; 27 match) as shown in Table 2. The total injury incidence for the hamstring muscles was 1.17 injuries/1000 hours (95% CI, 0.84–1.57). The injury incidence out of a total of 36,833 exposure hours was higher in matches 5.59 injuries/1000 hours (95% CI, 3.68–8.13) which is 11.2 times higher compared to training injury incidence, at 0.50 injuries/1000 hours (95% CI, 0.29–0.81). There were no differences in injury incidence concerning the players' position on the soccer field ($p = 0.258$). On average 27.9% of 143 players sustained at least one hamstring injury whilst three players suffered bilateral strains.

Training and match days lost due to HSIs were 643 in total. Injuries occurring on match days accounted for 431 lost training days, whereas injuries sustained in training accounted for 212 days of training absence. The re-injury rate of HSIs was 23%.

An independent-samples t-test was conducted to compare injured and non-injured players for age, anthropometry, exposure time, isokinetic performance, and functional parameters. As presented in Table 3, there was a significant difference in age between injured and non-injured players, with the injured players being older ($p < 0.001$). The body mass and BMI values were significantly higher among the injured compared to non-injured players (body mass, $p = 0.002$; BMI, $p = 0.002$). Height on the other hand did not differ between categories ($p = 0.505$). Based on Pearson Chi-Square test results, it can be noticed that there is a statistically significant difference in the passing rate of the NHST with a lower passing rate among the injured compared to non-injured players ($p = 0.001$), and a higher previous hamstring injury rate ($p = 0.023$), respectively. Independent samples t-test revealed no significant differences with

**Table 2. Injury incidence per training and match time exposure hours.**

|  | Exposure hours x 1000 | Injury incidence (95% CI) |
|---|---|---|
| Total HSIs incidence | 43 injuries / 36833 hours x 1000 | 1.17 (0.84–1.57) |
| Training | 16 injuries / 31998 hours x 1000 | 0.50 (0.29–0.81) |
| Match | 27 injuries / 4834 hours x 1000 | 5.59 (3.68–8.13) |

**Table 3. Characteristics according to age, anthropometric and physical fitness parameters between HSIs injured and non-injured players.**

| | Not injured (n = 103) | Injured (n = 40) | η² (95% CI) | p-Value (t-tests) |
|---|---|---|---|---|
| | Mean ± SD | Mean ±SD | | |
| Age (years) | 22.2 ± 3.9 | 26.1 ± 3.4 | 1.06 (-6.45 to -3.96) | <0.001*** |
| Body mass (kg) | 73.2 ± 6.6 | 77.0 ± 6.1 | 0.60 (-4.79 to 0.06) | 0.002** |
| BMI (kg/m²) | 22.6 ± 1.7 | 23.6 ± 1.5 | 0.62 (-1.68 to -0.46) | 0.002** |
| Height (cm) | 179.8 ± 5.6 | 180.5 ± 4.7 | 0.13 (-0.62 to 3.27) | 0.505 |
| NHST (no/yes, % of total) | 30/73 (21) | 24/16 (16.8) | (no: 1.14 to 1.31/ yes: 1.62 to 1.89) | 0.001** |
| Previous HSIs (no/yes, % of total) | 24/79 (23.3) | 17/23 (42.5) | (no: 0.15 to 0.32/ yes: 0.26 to 0.59) | 0.023* |
| Exposure time training (hours) | 225.4 ± 20.0 | 219.6 ± 24.9 | 0.26 (-6.92 to 8.88) | 0.154 |
| Exposure time match (hours) | 33.2 ± 9.0 | 35.3 ± 8.5 | 0.24 (-3.97 to 2.57) | 0.214 |
| Total exposure time (hours) | 258.6 ± 23.3 | 254.9 ± 28.7 | 0.14 (-8.85 to 9.42) | 0.432 |
| **Absolute concentric torque (60˚/s)** | | | | |
| Hamstring torque, dominant (Nm) | 129.4 ± 22.4 | 132.5 ± 23.3 | 0.14 (-6.10 to 10.45) | 0.467 |
| Hamstring torque, non-dominant (Nm) | 126.4 ± 22.2 | 130.6 ± 17.1 | 0.21 (-4.88 to 10.48) | 0.282 |
| Quads torque, dominant (Nm) | 224.3 ± 36.0 | 223.1 ± 30.2 | 0.04 (5.56 to 30.10) | 0.820 |
| Quads torque, non-dominant (Nm) | 226.8 ± 35.7 | 224.3 ± 35.8 | 0.07 (-3.73 to 22.23) | 0.707 |
| H/Q ratio, dominant (%) | 58.1 ± 7.7 | 59.9 ± 10.2 | 0.20 (7.03 to -0.95) | 0.245 |
| H/Q ratio, non-dominant (%) | 56.1 ± 8.3 | 59.8 ± 13.3 | 0.33 (-4.51 to 2.83) | 0.044* |
| **Absolute concentric torque (240˚/s)** | | | | |
| Hamstring torque, dominant (Nm) | 100.4 ± 20.2 | 105.9 ± 23.8 | 0.25 (-4.36 to 11.23) | 0.162 |
| Hamstring torque, non-dominant (Nm) | 98.2 ± 19.9 | 98.4 ± 21.6 | 0.01 (-5.50 to 9.40) | 0.969 |
| Quads torque, dominant (Nm) | 142. 2 ± 21.8 | 144.9 ± 23.8 | 0.12 (2.98 to 18.95) | 0.509 |
| Quads torque, non-dominant (Nm) | 142.1 ± 23.1 | 141.9 ± 21.1 | 0.01 (-6.21 to 10.30) | 0.945 |
| H/Q ratio, dominant (%) | 71.3 ± 12.9 | 73.4 ± 12.3 | 0.17 (-8.03 to 1.23) | 0.393 |
| H/Q ratio, non-dominant (%) | 69.5 ± 10.6 | 70.4 ± 16.6 | 0.06 (-3.51 to 5.67) | 0.711 |
| **Absolute eccentric torque** | | | | |
| Hamstring torque (ecc, 30˚/s) dominant (Nm) | 121.2 ± 45.3 | 127.0 ± 43.4 | 0.13 (-16.02 to 16.81) | 0.489 |
| Hamstring torque (ecc, 30˚/s) non-dominant (Nm) | 114.8 ± 44.9 | 118.2 ± 43.6 | 0.08 (-15.06 to 17.52) | 0.688 |
| Hamstring torque (ecc, 120˚/s) dominant (Nm) | 94.1 ± 48.9 | 101.8 ± 49.7 | 0.16 (-26.61 to 9.11) | 0.400 |
| Hamstring torque (ecc,120˚/s) non-dominant (Nm) | 92.1 ± 42.9 | 102.6 ± 44.7 | 0.24 (-21.01 to 10.83) | 0.195 |
| **Absolute eccentric torque** | | | | |
| Hamstring torque (ecc, 30˚/s) dominant (Nm) | 121.2 ± 45.3 | 127.0 ± 43.4 | 0.13 (-16.02 to 16.81) | 0.489 |
| Hamstring torque (ecc, 30˚/s) non-dominant (Nm) | 114.8 ± 44.9 | 118.2 ± 43.6 | 0.08 (-15.06 to 17.52) | 0.688 |
| Hamstring torque (ecc, 120˚/s) dominant (Nm) | 94.1 ± 48.9 | 101.8 ± 49.7 | 0.16 (-26.61 to 9.11) | 0.400 |
| Hamstring torque (ecc,120˚/s) non-dominant (Nm) | 92.1 ± 42.9 | 102.6 ± 44.7 | 0.24 (-21.01 to 10.83) | 0.195 |
| **Functional tests** | | | | |
| Countermovement jump (cm) | 44.9 ± 4.9 | 44.4 ± 4.9 | 0.10 (-0.96 to 2.62) | 0.581 |
| Illinois agility test (s) | 15.64 ± 0.62 | 15.65 ± 0.56 | 0.02 (-0.38 to 0.06) | 0.905 |
| 20 m dash (s) | 3.12 ± 0.16 | 3.14 ± 0.12 | 0.14 (-0.11 to 0.00) | 0.643 |
| 40 m dash (s) | 5.51 ± 0.27 | 5.57 ± 0.26 | 0.23 (-0.18 to 0.01) | 0.256 |
| Sit & reach (cm) | 28.3 ± 5.3 | 29.5 ± 5.5 | 0.22 (-0.70 to 3.20) | 0.239 |

Data are shown as mean ± standard deviation; differences between groups are calculated by independent samples t-test. Significant differences between hamstring injury group and not injured category are marked by asterisks (*p<0.05, **p<0.01, ***p<0.001). Differences in frequency distribution between groups were determined by Chi² test.

Abbreviations: Q (Quadriceps), H (Hamstrings); NHST (Nordic Hamstring Strength Test); "Dominant" refers to the kicking leg; η² refers to partial eta squared; CI confidence intervals.

respect to training, match and total exposure time between groups [(training, p = 0.154); (match, p = 0.214); (total, p = 0.432)].

When comparing dominant and non-dominant legs in terms of absolute concentric hamstring torque at 60˚/sec, no significant differences were observed between categories (dominant, p = 0.467; non-dominant, p = 0.282). Also, no cohort differences were revealed for the quadriceps torque at 60˚/sec in both legs (dominant, p = 0.820; non-dominant, p = 0.707). In addition, the H/Q ratio for the dominant leg was not significantly different between injured and non-injured players (p = 0.245), whereas significantly higher H/Q ratio values were observed among the injured players for the non-dominant leg (p = 0.044). However, there was no difference in absolute concentric torque for hamstrings, quadriceps and H/Q ratio between injured and non-injured players at 240˚/sec [(hamstrings dominant, p = 0.162); (hamstrings non-dominant, p = 0.969); (quadriceps dominant, p = 0.509); (quadriceps non-dominant, p = 0.945); (H/Q ratio dominant, p = 0.393); (H/Q ratio non-dominant, p = 0.711)]. Eccentric torque at 30˚/sec [(dominant, p = 0.489); (non-dominant, p = 0.688)], and at 120˚/sec [(dominant, p = 0.400); (non-dominant, p = 0.195)] was also not significantly different between categories. Further, no differences were found with respect to functional test parameters (countermovement jump, p = 0.581; Illinois agility test, p = 0.905, 20-meter dash, p = 0.643, 40-meter dash, p = 0.256, and sit-and-reach, p = 0.239).

Independent samples t-test did not reveal significant differences for relative concentric hamstring strength at 60˚/sec (dominant, p = 0.460); (non-dominant, p = 0.649), whereas for the quadriceps torque, the injured players attained significantly lower values for both legs compared to non-injured ones (dominant, p = 0.039; non-dominant, p = 0.025) (Table 4). In addition, the level of relative concentric hamstring and quadriceps strength produced at 240˚/sec was not significantly different between cohorts (p = 0.865). However, there was a tendency towards a better performance for the non-injured category (p = 0.055). On the other hand, the differences for relative eccentric hamstring strength were even smaller at 30˚/sec [(dominant:

**Table 4. Isokinetic values relative to body mass.**

|  | Not injured (n = 103) Mean ± SD | Injured (n = 40) Mean ± SD | η² (95% CI) | p-Value (t-tests) |
|---|---|---|---|---|
| **Relative concentric torque (60˚/s)** |  |  |  |  |
| Relative H torque, dominant (Nm/kg) | 1.77 ± 0.27 | 1.73 ± 0.34 | 0.13 (-0.01 to 0.20) | 0.460 |
| Relative H torque, non-dominant (Nm/kg) | 1.73 ± 0.27 | 1.70 ± 0.26 | 0.11 (0.01 to 0.20) | 0.649 |
| Relative Q torque, dominant (Nm/kg) | 3.07 ± 0.42 | 2.91 ± 0.40 | 0.39 (0.20 to 0.50) | 0.039* |
| Relative Q torque, non-dominant (Nm/kg) | 3.10 ± 0.42 | 2.92 ± 0.46 | 0.41 (0.10 to 0.38) | 0.025* |
| **Relative concentric torque (240˚/s)** |  |  |  |  |
| Relative H torque, dominant (Nm/kg) | 1.37 ± 0.24 | 1.38 ± 0.32 | 0.03 (-0.00 to 0.21) | 0.865 |
| Relative H torque, non-dominant (Nm/kg) | 1.34 ± 0.24 | 1.28 ± 0.29 | 0.22 (-0.02 to 0.20) | 0.220 |
| Relative Q torque, dominant (Nm/kg) | 1.94 ± 0.26 | 1.88 ± 0.32 | 0.21 (0.11 to 0.30) | 0.278 |
| Relative Q torque, non-dominant (Nm/kg) | 1.94 ± 0.26 | 1.85 ± 0.27 | 0.34 (-0.01 to 0.18) | 0.055 |
| **Relative eccentric torque** |  |  |  |  |
| Relative H torque, dominant (30˚/s, Nm/kg) | 1.67 ± 0.62 | 1.65 ± 0.56 | 0.03 (-0.16 to 0.28) | 0.903 |
| Relative H torque, non-dominant (30˚/s, Nm/kg) | 1.58 ± 0.63 | 1.54 ± 0.58 | 0.07 (-0.16 to 0.28) | 0.733 |
| Relative H torque, dominant (120˚/s, Nm/kg) | 1.30 ± 0.68 | 1.32 ± 0.64 | 0.03 (-0.31 to 0.18) | 0.849 |
| Relative H torque, non-dominant (120˚/s, Nm/kg) | 1.26 ± 0.58 | 1.33 ± 0.57 | 0.12 (-0.23 to 0.19) | 0.509 |

Data are shown as mean ± standard deviation; differences between groups are calculated by independent samples t-test. Significant differences between hamstring injury group and not injured category are marked by asterisks (*p<0.05, **p<0.01, ***p<0.001).

Abbreviations: Q (Quadriceps), H (Hamstrings); "Dominant" refers to the kicking leg

injured, p = 0.903); (non-dominant: injured, p = 0.733)], and 120˚/sec [(dominant: injured, p = 0.849); (non-dominant: injured, p = 0.509)], respectively.

Table 5 presents the results derived from the multiple linear regression models, which partially explain the variation for injury incidence as the outcome variable. Model 1 shows that age alone accounted for 18.1% of variation. Adding the NHST to the model (model 2) resulted in improved predictability which accounted for 21.3% of the variation of injury incidence. Besides age and the NHST, the addition of the previous HSIs further improved the determination effect (model 3), accounting for 23.8%. Finally, the addition of concentric hamstring strength of the dominant leg at 240˚/sec along with the parameters from the latter models increased the predictability of the hamstring injury to 25.9%. Body composition, isokinetic, functional tests (except for the dominant leg at 240˚/sec), and playing position did not improve the predictability outcome of hamstring injuries and were not reported as models by the statistical analysis.

## Discussion

The specific objectives of the present study were twofold. Firstly, it was attempted to prospectively investigate the hamstring injury profile and the incidence rate per 1000 hours exposure time during training and league matches in the Kosovo premier soccer league. The second aim was to assess if and which tests of a pre-season exercise test battery could be used to predict non-contact hamstring injuries.

In a previous epidemiological study among elite Kosovar soccer players, led by the group of the current study, it was shown that the overall injury incidence was approximately 20% lower compared to Western and Northern European soccer players [3]. However, when it comes to

**Table 5. Multiple linear regression models.**

|  | Models | β ± SE | (95%'CI) | p-Value | Adjusted R$^2$ (%) |
|---|---|---|---|---|---|
| **Hamstring injury rate** | 1 |  |  |  | 18.1 |
|  | Constant | - 0.79 ± 0.20 | -1.18 to -0.41 | <0.001 |  |
|  | Age | 0.05 ± 0.01 | 0.03 to 0.06 | <0.001 |  |
|  | 2 |  |  |  | 21.3 |
|  | Constant | -0.90 ± 0.2 | -1.30 to -0.52 | <0.001 |  |
|  | Age | 0.04 ± 0.01 | 0.02 to 0.06 | <0.001 |  |
|  | NHST | 0.20 ± 0.07 | 0.03 to 0.31 | 0.017 | 23.8 |
|  | 3 |  |  |  |  |
|  | Constant | -1.21 ± 0.24 | -1.68 to 0.73 | <0.001 |  |
|  | Age | 0.05 ± 0.01 | 0.03 to 0.07 | <0.001 |  |
|  | NHST | 0.25 ± 0.08 | 0.09 to 0.40 | 0.002 |  |
|  | Previous hamstring injury | -0.21 ± 0.10 | -0.41 to -0.01 | 0.035 |  |
|  | 4 |  |  |  | 25.9 |
|  | Constant | -1.53 ± 0.29 | -2.10 to -0.96 | <0.001 |  |
|  | Age | 0.05 ± 0.01 | 0.03 to 0.07 | <0.001 |  |
|  | NHST | 0.26 ± 0.08 | 0.11 to 0.42 | 0.001 |  |
|  | Previous hamstring injury | -0.21 ± 0.10 | -0.40 to -.01 | 0.037 |  |
|  | Con. HS (240˚/s–dominant leg) | 0.01 ± 0.01 | 0.00 to 0.01 | 0.049 |  |

Covariates are listed with parameter estimates and standard errors (β±SE) and with p-values. Con. HS (Concentric Hamstring Strength); NHST (Nordic Hamstring Strength Test).

hamstring injuries in particular, the injury rate in the current investigation was 1.17 injuries per 1000 hours (95% CI, 0.84 to 1.57). This is more or less similar to those reported by Ekstrand et al. (2016) [8] for high-level European soccer players [(1.20), (95% CI 1.14 to 1.26)] over 13 years. The results are also comparable regarding training injury rates [(0.50 injuries/ 1000 hours (95% CI, 0.29–0.81)] for Kosovo players versus [(0.51), (95% CI, 0.47 to 0.55)] for UEFA elite players. However, in the present investigation, the match injury rate tended to be higher [(5.59 injuries/1000hours), (95% CI, 3.68 to 8.13)] compared to professional UEFA players [(4.77), (95% CI, 4.49 to 5.06)]. This fact is surprising as the overall training and match load is substantially smaller for Kosovo players [3]. However, Kosovo is an underdeveloped state and the socioeconomic situation differs substantially compared to more developed countries and proper soccer infrastructure is still missing. Therefore, factors such as poor field conditions, decreased strength and conditioning status, as well as the lack of medical and sufficient coaching staff, which are all due to the lower professionalism, might have contributed to a higher injury incidence. Indeed, Iacovelli and colleagues (2013) found that when surface conditions of the natural playing soccer fields were poor, lower extremity injury rate was 2.61 times higher in comparison to normal field conditions [50]. In addition, at the time when the study was conducted, Kosovo was not a member of the International Federation of Football Associations (FIFA) or the Union of European Football Associations (UEFA), therefore the playing level might be considered less professional. Notably, injury occurrence and recurrence among amateur players, particularly during training, was reported to be higher in comparison with professionals [16, 51]. Furthermore, the physical preparedness of Kosovo players may be more precarious since to date strength and conditioning coaches licensed by the Kosovo Football Federation, FIFA or UEFA are still missing. In fact, strength and conditioning coaches might act as observers and correctors of exercise techniques aiming to develop a strong and resilient athlete and can implement scientific driven prevention programs [52].

In line with previous findings [2, 14, 25, 53–55], the current study also found a significant difference in the injury incidence between older-aged and previously injured players compared to younger and non-injured ones. Although HSIs are also well known to the younger athletic population, functional and structural changes particular to aging athletes diminish the ability to adapt to high levels of loading and make them more susceptible to certain pathological conditions that affect both muscles and tendons [56]. Previous injuries to the fascial system and injury-related interruption of the training process cause a significant loss of performance both in younger and older individuals. Greater decrements, however, are observed in rapid muscle force capacity in older individuals, who also appear to have an impaired capability to fully recover following injuries [57, 58]. A possible reason could be a higher co-contraction time of the antagonist lower limb muscles, a lower efficiency, and an uneven transmission of the forces within the fascicle compared to younger players [59, 60]. In addition, the passing rate of the NHST in the current study was significantly lower among injured compared to non-injured players. However, studies analyzing eccentric knee flexor strength found no association with an increased risk of future HSIs [23, 61]. This could, to some extent, explain the absence of differences in isokinetic eccentric hamstring torque in the present study between injured and non-injured players. Therefore, the lower passing rate of NHST might be attributed to other confounding factors rather than the lack of eccentric strength of the hamstrings. Yet, the effect of age, passing rate of NHST, and the previous injury history were shown to have a deterministic effect on HSIs in the current and previous investigations [31, 62], whereas except for concentric hamstring strength at 240˚/s for the dominant leg, the other isokinetic and functional risk factors did not show a significant predictability impact in the current study.

A study conducted by Bakken et al. (2018) revealed that greater quadriceps concentric peak torque at 60˚/sec correlated with the risk of overuse injuries [7]. However, van Dyk and peers

(2016) reported that quadriceps concentric and hamstring eccentric strength at 60˚/sec were low risk factors for HSIs [40]. In addition, Lee et al. (2018) suggested that measurements at lower isokinetic speeds (concentric, 60˚- and eccentric, 30˚/sec) might be useful preseason screening protocols to detect players who might be at risk of sustaining HSIs [63]. In the current investigation, the absolute quadriceps concentric peak torque was not different between injured and non-injured players, whereas the relative quadriceps torque of the same testing velocity was significantly lower among the injured players. However, the discrepancies between our findings and other author's findings may have occurred due to different protocols and testing methodology, rehabilitation programs after injury, and physical preparedness, which make it challenging for the results to be compared. Hamstring and quadriceps peak and relative torques at 240˚/sec were not different between cohorts in the current study. Similar results were also reported by Lee (2018) [63] suggesting that high isokinetic testing speeds may not be good predictors of HSIs.

Except for absolute concentric torque H/Q ratio where the peak torque production of the non-dominant leg of the previously injured category was slightly but significantly higher than the non-injured category, there were no other differences observed in H/Q ratio between cohorts in the present study. It has to be noted that the scientific evidence is not consistent when it comes to differences in H/Q ratio between dominant and non-dominant legs [64, 65]. However, the authors consider that this difference might be attributed to the compensatory adaptation of the non-dominant leg (balancing leg) which acts as the weight-bearing limb during the execution of the technical tasks against the dominant one. Based on cut-off points established by Croisier et al. (2008) who suggest that an H/Q ratio lower than 0.55% would classify players at increased risk of suffering HSIs [43], both categories were not classified at increased risk in the current study, as the values were higher than Croisier's cut-off points (all > 0.55%). This was also shown in other papers with authors reporting that H/Q ratio may not be considered a reliable risk factor at all [40, 66]. However, other investigations [43, 67, 68] have identified the H/Q ratio as a reliable risk factor for HSIs, yet with disputes concerning the interpretation of this parameter. Two of the largest studies concerning the HSIs prediction in soccer players found mismatching results concerning the H/Q ratio [40, 43] making it difficult to determine whether this parameter could be considered a valid and reliable risk factor.

The present study revealed no difference in functional and motoric tests between injured and non-injured players during acceleration, maximal running speed, agility, and counter-movement jump tests. The ability to perform and to sustain multiple brief high-intensity intermittent activities and continuous changes of direction during match play is a key attribute for soccer performance. However, this performance profile has shown to result in temporary neuromuscular fatigue specifically, which might increase the risk of HSIs at the end of each half of soccer matches [15, 69]. In fact, it has been previously shown that fatigue can affect neuromuscular control and multi-muscle coordination patterns resulting in altered lower limb biomechanics as well as running and movement kinematics [70–73], factors which could be related to hamstring injuries.

Insufficient neuromuscular and kinetic chain control has been proposed as a risk factor for non-contact injuries [70]. Supporting earlier findings from Chumanov et al. (2007) [74–76], Schuermans et al. (2016, 2017) and more recently (2018) [20] observed that deficient core stability and lumbopelvic control during sprinting, specifically an inefficient posterior chain muscle recruitment pattern, might result in an increased hamstring injury risk. However, it has been criticized that prospective risk factor studies assessing the effect of lumbopelvic motion on hamstring injury rates are widely missing [19]. Additionally, previous injuries, either hamstring injuries or injuries to other body parts, might influence neuromuscular coordination and control. Malliaropoulos et al. (2018) [77] recently reported a statistically significant

interdependence of HSIs with previous traumatic ankle ligament injuries in track and field athletes. Therefore, a prior injury to the lower limbs could be a predisposing factor for future HSIs. Our study population showed to have a high recurrence rate of HSIs, confirming this association.

At this point, it is important to note that testing in the current study was performed in a non-fatigued state. However, injury risk in general and hamstring risk more specifically cannot be captured by an assessment performed in a rested state but ideally should be conducted in a fatigued state creating an ecologically valid environment. This indicates, that injury risk assessment should not be performed simultaneously with pre-season exercise or physiological assessments or strength and conditioning screening. These tests aim to provide information about the actual fitness status of the players or to determine the progression and effectiveness of a training program. However, assessments focusing on identifying players at risk of HSIs need to have an entirely different emphasis which is specifically tailored to detecting the risk factors of HSIs.

Successful identification of injury predictors or dysfunction forms the basis of effective preventive measures. However, such appropriate injury prevention strategies can only be developed based on a comprehensive understanding of potential prognostic factors and based on valid and reliable tests for predicting hamstring injuries [6]. Despite a novel and comprehensive methodology including a homogenous study population, and a vast number of field and laboratory tests, some potential limitations are worth noting. Notably, a longer monitoring period could provide a better prediction depiction of HSIs. Further, different diagnostic, prevention, and rehabilitation protocols implemented by the teams' staff could have influenced the results. Therefore, further research including standardized diagnostic, preventive, and load monitoring measures are of utmost importance in order to better predict HSIs in soccer players.

## Conclusions

The main findings of this prospective study revealed that a battery of laboratory and field-based tests performed during pre-season to determine performance-related skills is of limited diagnostic conclusiveness, preventing the detection of players at risk of future hamstring injuries. Subsequently, appropriate individualized injury prevention strategies generated to reduce the perceived risk need to be created. Age, NHST, previous injury history, and concentric hamstring strength at 240˚/sec resulted to be the best prediction factors of HSIs. Hence, it is obvious that hamstring injuries are a multifactorial problem involving the interaction of various confounding components that heighten the risk of hamstring injuries, posing the demands for further research in detecting the best predicting model of HSIs.

## Supporting information

**S1 File.**
(SAV)

## Acknowledgments

The work was performed in collaboration with the University of Pristina, Medical Faculty, Department of Physiotherapy. Special thanks is dedicated to Feim Gashi, MSc, and Besim Ademi MSc for their valuable help with data acquisition. Furthermore, we gratefully acknowledge the compliance of players, medical personnel, and contact persons from the respective soccer clubs involved in the study. Finally, we appreciate the cooperation of the Kosovo Football Federation.

## Author Contributions

**Conceptualization:** Ismet Shalaj, Harald Tschan, Faton Tishukaj.

**Data curation:** Ismet Shalaj, Masar Gjaka, Barbara Wessner, Faton Tishukaj.

**Formal analysis:** Ismet Shalaj, Masar Gjaka, Barbara Wessner, Harald Tschan, Faton Tishukaj.

**Investigation:** Ismet Shalaj, Masar Gjaka, Harald Tschan, Faton Tishukaj.

**Methodology:** Ismet Shalaj, Harald Tschan, Faton Tishukaj.

**Project administration:** Ismet Shalaj, Masar Gjaka, Faton Tishukaj.

**Software:** Faton Tishukaj.

**Supervision:** Norbert Bachl, Faton Tishukaj.

**Validation:** Faton Tishukaj.

**Visualization:** Faton Tishukaj.

**Writing – original draft:** Ismet Shalaj, Harald Tschan, Faton Tishukaj.

**Writing – review & editing:** Ismet Shalaj, Masar Gjaka, Harald Tschan, Faton Tishukaj.

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
