## [Decision Letter · Decision Letter 0]

26 Nov 2019

PONE-D-19-20247

Potential prognostic factors for hamstring muscle injury in elite male soccer players: a prospective study

PLOS ONE

Dear Univ.-Prof. Dr. Tschan,

Thank you for submitting your manuscript to PLOS ONE. After careful consideration, we feel that it has merit but does not fully meet PLOS ONE’s publication criteria as it currently stands. Therefore, we invite you to submit a revised version of the manuscript that addresses the points raised during the review process.

We would appreciate receiving your revised manuscript by Jan 10 2020 11:59PM. To enhance the reproducibility of your results, we recommend that if applicable you deposit your laboratory protocols in protocols.io, where a protocol can be assigned its own identifier (DOI) such that it can be cited independently in the future. For instructions see: http://journals.plos.org/plosone/s/submission-guidelines#loc-laboratory-protocols

We look forward to receiving your revised manuscript.

Kind regards,

Yumeng Li

Academic Editor

PLOS ONE

Journal Requirements:

2. We note that your study is closely related to the following publication, on which you are an author:

https://bmcmusculoskeletdisord.biomedcentral.com/articles/10.1186/s12891-016-1202-9

Although you have briefly cited the above study in the methods section of your article, we feel that the scientific rationale of the current study and the contribution that it makes to the field should be justified better. Therefore, please cite and discuss the above study in the introduction, methods and discussion sections of your manuscript, clarifying how the present work is related to the previously published paper.

Please note that our second publication criterion states that "If a submitted study replicates or is very similar to previous work, authors must provide a sound scientific rationale for the submitted work and clearly reference and discuss the existing literature. Submissions that replicate or are derivative of existing work will likely be rejected if authors do not provide adequate justification." http://www.plosone.org/static/publication.action#results.

Thank you for your attention to this request.

Additional Editor Comments:

Based on reviewers' comments, some major revisions are needed before publication. Also, please check grammar/spelling throughout the paper.

Reviewers' comments:

Reviewer's Responses to Questions

**Comments to the Author**

1. Is the manuscript technically sound, and do the data support the conclusions?

Reviewer #1: Yes

Reviewer #2: Partly

Reviewer #3: Partly

Reviewer #4: Yes

2. Has the statistical analysis been performed appropriately and rigorously? 

Reviewer #1: Yes

Reviewer #2: N/A

Reviewer #3: No

Reviewer #4: Yes

3. Have the authors made all data underlying the findings in their manuscript fully available?

Reviewer #1: Yes

Reviewer #2: Yes

Reviewer #3: No

Reviewer #4: No

4. Is the manuscript presented in an intelligible fashion and written in standard English?

Reviewer #1: Yes

Reviewer #2: No

Reviewer #3: Yes

Reviewer #4: Yes

5. Review Comments to the Author

Reviewer #1: Summary

This is a comprehensive study that has investigated the incidence and characteristics of hamstrings injury in professional level soccer players. This is certainly an area that requires investigation given the incidence in hamstring injuries in this population and the impact on player health and wellness, and the revenue associated with players missing time on the field. Clearly, if definitive prognostic markers could be identified and prevention measures could be strategically employed to reduce the risk of hamstring injury, this would have enormous benefit in this field. Thus, the study and the overall area of research was definitely warranted and the authors did a comprehensive job at evaluating their data and attempting to identify these areas. Unfortunately, this study could not identify any battery of tests and field based tests that could conclusively diagnose increased risk of hamstring injury. Based on these findings, this study confirms that hamstring injuries are a multifaceted problem with numerous variables that interact to contribute towards injuries.

Minor Points

There was a difference in the H/Q absolute concentric torque on the non-dominant leg and this was not addressed in the discussion section. This reviewer suggests the authors attempt to explain this finding perhaps in the context of a compensatory adaptation of the non-dominant leg in an attempt to potentially protect against' injury (perhaps address at lines 387-394).

Grammatical (Gr) and Spelling (Sp)

line 116-117-Gr-"might be a safe treat?"

line 124-Gr-"ex ante"

line 126-Sp"suing" should be using

line 129-Gr-insert "a" before weekly

line 136- missing close bracket after m2

line 324-dash between two-fold

line 372-Sp-grater to greater

Table 4 Relative Q torque, dominant (60deg/s) should be 3.07 and not 30.7

Reviewer #2: General comment

The authors provided us with data about factors that can explain hamstring injuries in high level soccer teams. There is a good attempt to congregate in a single study a large number of field and lab tests that are able to predict hamstring injury. However, the time and the way how the intervention was implemented leave us some doubts about the real novelty of the study. I leave to the authors some comments for consideration.

Specific comments

The paper needs some work to improve the English sentence structure. There are several grammar errors. Please seek some further assistance on overall manuscript.

Are HIS determined by team level and division positioning? I think that you should refer to this in your introduction section.

Line 91: please provide the age threshold that you refer as being a non-modifiable risk factor for HSI.

Do you think that using a sample from Kosovo soccer league can explain HSI in a high standard level such as Champions League teams? Are Kosovo teams with good and enough technical teams and/or medical support to prescribe the most appropriate tests and/or methods for injury prevention?

I understand that you used data from 2013-2014 soccer season. But, from those previous days until now there was a large increase in scientific evidence regarding soccer injuries, with HSI are included. Don’t you think that such kind of experimental data can be outdated? I refer to this aspect because, during this time period, the training methods were changing as well. Please reflect on that.

Please add a hypothesis after the aim of your study.

What were the inclusion criteria of your subjects?

Please provide detailed information about the testing distribution. Where those tests performed in a single day? How many times where performed in a single season?

Authors should report the confidence limit for statistical significance. Also should consider to add an effect size measure to strengthen their statistical analysis.

From your results I may conclude that HSI are most determined by non-match factors (e.g. age, body mass, Nordic test and previous injuries). They cannot avoid the “age factor” at some point of their career. You didn’t dissected the different components of body mass, not allowing to understand how those 70 kg where attributed to lean and fat mass. At the end, the higher hamstring strength given by Nordic exercise, and the previous injuries were the factors with differences. But, from my perspective, this is not new when considering data from previous studies.

At the end I see your study as descriptive of the HSI occurrence in a single soccer league. In my opinion one more interesting approach would be to define a deterministic model that would identify the most important factors for HSI occurrence in the injury cohort, giving to us the exact contribution for HSI appearance.

Reviewer #3: This is a valuable observational prospective cohort study that examined 143 elite male soccer players. However, there are some issues that needs to be addressed:

Line 203-204: The sentence has a grammatical error, rewrite the sentence.

Line 249-254: Specify the units of the parameters

In the statistical analysis: the researchers only compared parameters between injured and uninjured soccer players and the factors that had significant differences between the two groups have reported as the main parameters for predicting injuries. However, the regression analysis can show the most important factors that can predict the occurrence of HIS. It recommends to do statistical analysis by this method (Regression analysis) and report the most important factors of HIS injuries and discuss these parameters.

In the results part: there are no need to rewrite all the results, as they are depicted in the table 3 and 4. It suggests to add one more column into this table and write the value of the t-test in this column and delete the repeated results.

In the results part: Specify the effect size and observed power of t- test analysis in two more column of tables.

Reviewer #4: The authors investigate the incidence and characteristics of hamstring injuries in elite male football players in the Kosovo, and found that laboratory and field-based tests are limited in prospectively determine the risk of hamstring injuries. The study design, perform, data preparation, and manuscript writing were good. However, I can’t get any new findings or from this paper, it seems that this manuscript just repeat the previous study. Although there was no investigation in the Kosovo football players for hamstring injuries risk and this study found passing rate of NHST, H/Q ratio for prospectively determine the risk of hamstring injuries were different from the previous studies, it’s hard to use these new findings such as for other players. Thus, it’s not recommened to publish this manuscript.

6. PLOS authors have the option to publish the peer review history of their article (what does this mean?). If published, this will include your full peer review and any attached files.

Reviewer #1: No

Reviewer #2: No

Reviewer #3: No

Reviewer #4: Yes: Xi Chen

---

## [Author Response · Author response to Decision Letter 0]

14 Jun 2020

Journal Requirements:

2. We note that your study is closely related to the following publication, on which you are an author:

https://bmcmusculoskeletdisord.biomedcentral.com/articles/10.1186/s12891-016-1202-9

Although you have briefly cited the above study in the methods section of your article, we feel that the scientific rationale of the current study and the contribution that it makes to the field should be justified better. Therefore, please cite and discuss the above study in the introduction, methods and discussion sections of your manuscript, clarifying how the present work is related to the previously published paper.

Please note that our second publication criterion states that "If a submitted study replicates or is very similar to previous work, authors must provide a sound scientific rationale for the submitted work and clearly reference and discuss the existing literature. Submissions that replicate or are derivative of existing work will likely be rejected if authors do not provide adequate justification." http://www.plosone.org/static/publication.action#results.

Thank you for your attention to this request.

The authors would like to thank the Editor and the Reviewers for their positive and thought-provoking comments, which have improved the overall quality of the paper. The manuscript has been changed based on the reviewer’s comment and our point-by-point responses, highlighting the changes made are outlined below. In the text of the revised manuscript the color highlights the modified sentences. 

The paper that the Editor is referring to has dealt with the epidemiology of the overall injuries in soccer players with no particular emphasis neither on hamstring injuries (HSIs), nor in attempting to predict HSIs. The first publication did not provide data concerning the assessed physical fitness parameters, and the data collection typology was completely different as described in the methodology of both articles. 

As requested by the Editor, a reasonable justification of the first publication can be found in the introduction (page 3, lines 64-65) as follows: 

In this regard, country-specific differences and potential injury mediators have been reported.

We thank the Editor for raising this issue and we have uploaded the anonymized data set entitled as Supporting Information. 

Comments to the Author

1. Is the manuscript technically sound, and do the data support the conclusions?

Reviewer #1: Yes

Reviewer #2: Partly

Reviewer #3: Partly

Reviewer #4: Yes

2. Has the statistical analysis been performed appropriately and rigorously?

Reviewer #1: Yes

Reviewer #2: N/A

Reviewer #3: No

Reviewer #4: Yes

3. Have the authors made all data underlying the findings in their manuscript fully available?

Reviewer #1: Yes

Reviewer #2: Yes

Reviewer #3: No

Reviewer #4: No

4. Is the manuscript presented in an intelligible fashion and written in standard English?

Reviewer #1: Yes

Reviewer #2: No

Reviewer #3: Yes

Reviewer #4: Yes

5. Review Comments to the Author

Reviewer #1: Summary

This is a comprehensive study that has investigated the incidence and characteristics of hamstrings injury in professional level soccer players. This is certainly an area that requires investigation given the incidence in hamstring injuries in this population and the impact on player health and wellness, and the revenue associated with players missing time on the field. Clearly, if definitive prognostic markers could be identified and prevention measures could be strategically employed to reduce the risk of hamstring injury, this would have enormous benefit in this field. Thus, the study and the overall area of research was definitely warranted and the authors did a comprehensive job at evaluating their data and attempting to identify these areas. Unfortunately, this study could not identify any battery of tests and field based tests that could conclusively diagnose increased risk of hamstring injury. Based on these findings, this study confirms that hamstring injuries are a multifaceted problem with numerous variables that interact to contribute towards injuries.

Minor Points

There was a difference in the H/Q absolute concentric torque on the non-dominant leg and this was not addressed in the discussion section. This reviewer suggests the authors attempt to explain this finding perhaps in the context of a compensatory adaptation of the non-dominant leg in an attempt to potentially protect against' injury (perhaps address at lines 387-394).

The authors thank Reviewer 1 for his/her comments. Accordingly, we have inserted the discussion concerning the H/Q absolute concentric torque of the non-dominant leg (page 19, lines 388-395) as follows: 

Except for absolute concentric torque H/Q ratio where the peak torque production of the non-dominant leg of the previously injured category was slightly but significantly higher, there were no other differences observed in H/Q ratio between cohorts in the present study. It has to be noted that the scientific evidence is not consistent when it comes to differences in H/Q ratio between dominant and non-dominant legs (64, 65). However, the authors consider that this difference might be attributed to the compensatory adaptation of the non-dominant leg (balancing leg) which acts as a weight-bearing limb during the execution of the technical tasks against the dominant one.

Grammatical (Gr) and Spelling (Sp)

line 116-117-Gr-"might be a safe treat?"

line 124-Gr-"ex ante"

line 126-Sp"suing" should be using

line 129-Gr-insert "a" before weekly

line 136- missing close bracket after m2

line 324-dash between two-fold

line 372-Sp-grater to greater

The authors thank the Reviewers 1 for his/her valuable comments and suggestions and we have made all Grammatical and Spelling corrections accordingly. 

Table 4 Relative Q torque, dominant (60deg/s) should be 3.07 and not 30.7

The authors thank the Reviewers 1 for his/her valuable observation and the correction in Table 4 has been made. 

Reviewer #2: General comment

The authors provided us with data about factors that can explain hamstring injuries in high level soccer teams. There is a good attempt to congregate in a single study a large number of field and lab tests that are able to predict hamstring injury. However, the time and the way how the intervention was implemented leave us some doubts about the real novelty of the study. I leave to the authors some comments for consideration.

Specific comments

The paper needs some work to improve the English sentence structure. There are several grammar errors. Please seek some further assistance on overall manuscript.

The authors thank the Reviewers 2 for his/her valuable comments and suggestions and we have checked the improved overall English sentence structure of the manuscript. 

Are HIS determined by team level and division positioning? I think that you should refer to this in your introduction section.

The authors thank the Reviewers 2 for his/her valuable comment which have contributed to improve the quality of the manuscript, and have added the information required (page 3, lines 75-76) as follows: 

On the other hand, a lower injury rate of HSIs has been reported among lower division soccer clubs (16).

Line 91: please provide the age threshold that you refer as being a non-modifiable risk factor for HSI.

Thank you Reviewer 2 for your suggestion. Unfortunately, the authors could not find a paper determining the age threshold. However, we have included the justification concerning the age factor (page 4, lines 94-96) as follows: 

With respect to non-modifiable risk factors previous hamstring injury and increasing age have shown to increase the odds to suffer a HSI by 11.6, and 1.4, respectively (31).

Do you think that using a sample from Kosovo soccer league can explain HSI in a high standard level such as Champions League teams? Are Kosovo teams with good and enough technical teams and/or medical support to prescribe the most appropriate tests and/or methods for injury prevention?

Thank you Reviewer 2 for your comments concerning this issue. The authors think that we have extensively addressed this issue in the “discussion” section of the manuscript (pages 16-17, lines 334-350) as follows: 

Nevertheless, Kosovo is a state under development and the socioeconomic situation differs substantially compared to developed countries and proper soccer infrastructure is still missing. Therefore, factors such as field conditions, professional level and strength and conditioning status might have contributed to a higher injury incidence. Indeed, Iacovelli and colleagues (2013) found that when surface conditions of the natural playing soccer fields were abnormal, lower extremity injury rate was 2.61 times higher in comparison to a normal field condition (50). In addition, at the time when the study was conducted, Kosovo was not a member of International Federation of Football Associations (FIFA) and Union of European Football Associations (UEFA), therefore the playing level might be considered less-professional. Notably, injury occurrence and recurrence among amateur players, particularly during training, was reported to be higher in comparison with professional ones (16, 51). Furthermore, the physical preparedness of Kosovo players may be more precarious since, to date, strength and conditioning coaches licensed by Kosovo Football Federation, FIFA or UEFA are still missing. In fact, strength and conditioning coaches might act as observers and correctors of exercise techniques aiming to develop a strong and resilient athlete and can implement scientific driven prevention programs (52).

I understand that you used data from 2013-2014 soccer season. But, from those previous days until now there was a large increase in scientific evidence regarding soccer injuries, with HSI are included. Don’t you think that such kind of experimental data can be outdated? I refer to this aspect because, during this time period, the training methods were changing as well. Please reflect on that.

The authors thank Reviewer 2 for this important comment. The authors believe that since the results of the present manuscript are in line with recent publications, the information and findings of the current paper could play a role when hamstring injury prediction is concerned. It is widely accepted that the time lag between evidence being produced and its use in practice takes almost 2 decades. 

We leave to the Reviewer 2 for his/her discretion an article dealing with translational research that supports the author’s argument. 

Morris ZS, Wooding S, Grant J. The answer is 17 years, what is the question: Understanding time lags in translational research. J R Soc Med. 2011;104(12):510–20.

Please add a hypothesis after the aim of your study.

Thank you Reviewer 2 for your suggestion. We have included the hypothesis after the aims of the study accordingly (page 5, lines 110-112) as follows:

For the present study, it was hypothesized that players having one or more risk factors would be more prone to suffer HSIs.

What were the inclusion criteria of your subjects?

The authors thank the Reviewers 2 for his/her valuable comment which have contributed to improve the quality of the manuscript, and have added the information required (page 5, lines 125-127) as follows: 

Players suffering from acute lower limb injuries or recovering from recent surgical interventions (within the last 12 months) were excluded

Please provide detailed information about the testing distribution. Where those tests performed in a single day? How many times where performed in a single season?

Thank you Reviewer 2 for your suggestion. The authors have inserted the testing distribution information (page 7, lines 160-164) as follows: 

Isokinetic torque measurements have been performed on a separate day within the same week as other tests of the test battery consisting of the sit and reach, and Nordic hamstring strength test performed in the listed order. Finally, the countermovement jump, speed, and agility tests were performed on the last testing day. All tests were performed only before season kick-off.

Authors should report the confidence limit for statistical significance. Also should consider to add an effect size measure to strengthen their statistical analysis.

Thank you Reviewer 2 and 3 for this important suggestion. The authors have inserted the confidence intervals and effect size accordingly which are inserted in Table 3 and 4. 

From your results I may conclude that HSI are most determined by non-match factors (e.g. age, body mass, Nordic test and previous injuries). They cannot avoid the “age factor” at some point of their career. You didn’t dissected the different components of body mass, not allowing to understand how those 70 kg where attributed to lean and fat mass. At the end, the higher hamstring strength given by Nordic exercise, and the previous injuries were the factors with differences. But, from my perspective, this is not new when considering data from previous studies.

Thank you Reviewer 2 for this observation and suggestion. When we performed the regression analysis, body mass was not identified as a risk factor. Since we did not perform body composition analysis, but only calculated the BMI, we did not want to go into more detail concerning this issue. 

At the end I see your study as descriptive of the HSI occurrence in a single soccer league. In my opinion one more interesting approach would be to define a deterministic model that would identify the most important factors for HSI occurrence in the injury cohort, giving to us the exact contribution for HSI appearance.

The authors thank Reviewer 2 and 3 for their valuable comments which have contributed to improve the quality of statistical analysis strengthening the meaning of the outcomes. The authors have performed the multiple-linear regression analysis consequently (page 15, lines 305-315) as follows: 

Table 5 presents the results derived from the multiple linear regression models, which partially explain the variation for injury incidence as the outcome variable. Model 1 shows that age alone accounted for 18.1%. When Nordic hamstring test was added to the model (model 2) yielded improved predictability which accounted for 21.3% of the variation of injury incidence. Besides, age, Nordic hamstring test, and the addition of the previous hamstring strain injury further improved the determination effect (model 3) accounting for 23.8%. Finally, the addition of concentric hamstring strength of the dominant leg at 240º/sec along with the parameters from the latter model increased the predictability of the hamstring injury to 25.9%. Body composition, isokinetic, functional tests (except for the dominant leg at 240º/sec), and playing position, did not improve the predictability outcome of hamstring injuries and were not reported as models by the statistical analysis.

Reviewer #3: This is a valuable observational prospective cohort study that examined 143 elite male soccer players. However, there are some issues that needs to be addressed:

Line 203-204: The sentence has a grammatical error, rewrite the sentence.

Thank you Reviewer 2 for your suggestion. The authors have rewritten the sentence (page 9, lines 219-220) as follows: 

Descriptive statistics, such as age, anthropometric, and physical performance tests were used for baseline characteristics and means ± SD were calculated.

Line 249-254: Specify the units of the parameters

Thank you Reviewer 2 for your suggestion. We have specified the units of the parameters accordingly as presented in Table 2. 

In the results part: there are no need to rewrite all the results, as they are depicted in the table 3 and 4. It suggests to add one more column into this table and write the value of the t-test in this column and delete the repeated results.

The authors thank Reviewer 3 for his/her suggestion which has improved the “results” section of the manuscript. The authors have made the suggested changes accordingly (pages11-14, lines 266-303) as follows:

An independent-samples t-test was conducted to compare injured and non-injured players for age, anthropometry, exposure time, isokinetic performance and functional parameters. As presented in Table 3, there was a significant difference in age between injured and non-injured players (p < 0.001). The body mass and BMI values were significantly higher among the injured compared to non-injured players (body mass, p = 0.002; BMI, p = 0.002), respectively. Height on the other hand did not differ between categories (p = 0.505). Besides, based on Pearson Chi-Square test results, it can be noticed that there is a statistically significant difference in the passing rate of Nordic Hamstring test results with a lower passing rate among the injured compared to non-injured players (p = 0.001), and a higher previous hamstring injury rate (p = 0.023), respectively. On the other hand, independent samples t-test revealed no significant differences with respect to training, match and total exposure time between groups [(training, p = 0.154); (match, p = 0.214); (total, p = 0.432)].

When comparing dominant and non-dominant legs in absolute concentric hamstring torque at 60º/sec, no significant differences were observed between categories (dominant, p = 0.467; non-dominant, p = 0.282). No cohort differences were also revealed for the quadriceps torque at 60º/sec in both legs (dominant, p = 0.820); non-dominant, p = 0.707). Adding, the hamstring/quadriceps (H/Q) ratio for the dominant leg was not significantly different between injured and non-injured players (p = 0.245), whereas significantly higher H/Q ratio values were observed among the injured players for the non-dominant leg (p = 0.044). On the other hand, there was no difference for absolute concentric torque for hamstrings, quadriceps and H/Q ratio between injured and non-injured players at 240º/sec [(hamstrings dominant, p = 0.162); (hamstrings non-dominant, p = 0.969); (quadriceps dominant, p = 0.509); (quadriceps non-dominant, p = 0.945); (H/Q ratio dominant, p = 0.393); (H/Q ratio non-dominant, p = 0.711)]. Eccentric torque at 30º/sec [(dominant, p = 0.489); (non-dominant, p = 0.688)], and at 120º/sec [(dominant, p = 0.400); (non-dominant, p = 0.195)] was also not significantly different between categories. Further, no differences were found with respect to functional test parameters (countermovement jump, p = 0.581); Illinois agility test, p = 0.905, 20-meter dash: injured, p = 0.643, 40-meter dash, p = 0.256, and sit-and-reach, p = 0.239).

Independent samples t-test did not reveal significant differences for relative concentric hamstring strength at 60º/sec (dominant, p = 0.460); (non-dominant, p = 0.649), whereas for the quadriceps torque, the injured players attained significantly lower values for both legs compared to non-injured ones (dominant, p = 0.039); (non-dominant, p = 0.025), respectively (Table 4). In addition, the level of relative concentric hamstring and quadriceps strength produced at 240º/sec was not significantly different between cohorts (p = 0.865). However, there was a tendency for a better performance for the non-injured category (p = 0.055). On the other hand, the differences for relative eccentric hamstring strength were even narrower at 30º/sec [(dominant: injured, p = 0.903); (non-dominant: injured, p = 0.733)], and 120º/sec [(dominant: injured, p = 0.849); (non-dominant: injured, p = 0.509)], respectively.

---

## [Decision Letter · Decision Letter 1]

31 Jul 2020

PONE-D-19-20247R1

Potential prognostic factors for hamstring muscle injury in elite male soccer players: a prospective study

PLOS ONE

Dear Dr. Tschan,

Thank you for submitting your manuscript to PLOS ONE. After careful consideration, we feel that it has merit but does not fully meet PLOS ONE’s publication criteria as it currently stands. Therefore, we invite you to submit a revised version of the manuscript that addresses the points raised during the review process.

We look forward to receiving your revised manuscript.

Kind regards,

Yumeng Li

Academic Editor

PLOS ONE

Additional Editor Comments (if provided):

Most reviewers commented that this paper lacks of novelty and readers cannot get sufficient new information. The authors should provide more information about the novelty and significance of the present study.

Reviewers' comments:

Reviewer's Responses to Questions

**Comments to the Author**

1. If the authors have adequately addressed your comments raised in a previous round of review and you feel that this manuscript is now acceptable for publication, you may indicate that here to bypass the “Comments to the Author” section, enter your conflict of interest statement in the “Confidential to Editor” section, and submit your "Accept" recommendation.

Reviewer #2: All comments have been addressed

Reviewer #3: All comments have been addressed

2. Is the manuscript technically sound, and do the data support the conclusions?

Reviewer #2: Yes

Reviewer #3: Yes

3. Has the statistical analysis been performed appropriately and rigorously? 

Reviewer #2: Yes

Reviewer #3: Yes

4. Have the authors made all data underlying the findings in their manuscript fully available?

Reviewer #2: Yes

Reviewer #3: Yes

5. Is the manuscript presented in an intelligible fashion and written in standard English?

Reviewer #2: No

Reviewer #3: Yes

6. Review Comments to the Author

Reviewer #2: The authors provided reasonable changes according with the reviewers opinion. Despite that I still maintain my reserves about the novelty of your study. In this sense I just have minor suggestions.

- At the end you should add a limitations' section to better describe the flaus of the intervention;

- Please make your conclusions more clearly for the reader. If the Nordic exercise performance predicts better the hamstrings injury incidence, this should be one of your key points.

Reviewer #3: (No Response)

7. PLOS authors have the option to publish the peer review history of their article (what does this mean?). If published, this will include your full peer review and any attached files.

Reviewer #2: No

Reviewer #3: **Yes: **Ali Abbasi

---

## [Author Response · Author response to Decision Letter 1]

9 Sep 2020

Comments to the Author

The authors would like to thank the Editor and Reviewer 2 for his/her positive and thought-provoking comments, which have improved the overall quality of the paper. The manuscript has been changed based on the reviewer’s comments and our point-by-point responses, highlighting the changes made are outlined below. In the text of the revised manuscript the color highlights the modified sentences. 

5. Is the manuscript presented in an intelligible fashion and written in standard English?

The authors thank the Editor and Reviewer 2 for their request to further improve the English language in the manuscript and the corrections were made accordingly. 

6. Review Comments to the Author

Reviewer #2: The authors provided reasonable changes according with the reviewers opinion. Despite that I still maintain my reserves about the novelty of your study. In this sense I just have minor suggestions.

- At the end you should add a limitations' section to better describe the flaus of the intervention;

- Please make your conclusions more clearly for the reader. If the Nordic exercise performance predicts better the hamstrings injury incidence, this should be one of your key points.

The authors thank Reviewer 2 for his/her valuable comments concerning the limitations of the study and the respective changes can be found in: page 21; lines 448-455. 

Despite a novel and comprehensive methodology including a homogenous study population, and a vast number of field and laboratory tests, some potential limitations are worth noting. Notably, a longer monitoring period could provide a better prediction depiction of HSIs. Further, different, diagnostic, prevention, and rehabilitation protocols implemented by the teams’ staff could have influenced the results. Therefore, further research including standardized diagnostic and preventive, and load monitoring measures are of utmost importance in order to better predict HSIs in soccer players. 

The authors thank Reviewer 2 for his/her valuable comments with respect to the conclusions section and have improved the section as follows: page 21-22; lines 462-463.

Age, NHST, previous injury history, and concentric hamstring strength at 240°/sec resulted to be the best prediction factors of HSIs.

---

## [Decision Letter · Decision Letter 2]

9 Oct 2020

Potential prognostic factors for hamstring muscle injury in elite male soccer players: a prospective study

PONE-D-19-20247R2

Dear Dr. Tschan,

We’re pleased to inform you that your manuscript has been judged scientifically suitable for publication and will be formally accepted for publication once it meets all outstanding technical requirements.

Kind regards,

Yumeng Li

Academic Editor

PLOS ONE

Additional Editor Comments (optional):

All comments have been successfully addressed. The paper could be accepted in its current form.

Reviewers' comments:

Reviewer's Responses to Questions

**Comments to the Author**

1. If the authors have adequately addressed your comments raised in a previous round of review and you feel that this manuscript is now acceptable for publication, you may indicate that here to bypass the “Comments to the Author” section, enter your conflict of interest statement in the “Confidential to Editor” section, and submit your "Accept" recommendation.

Reviewer #2: All comments have been addressed

2. Is the manuscript technically sound, and do the data support the conclusions?

Reviewer #2: Yes

3. Has the statistical analysis been performed appropriately and rigorously? 

Reviewer #2: N/A

4. Have the authors made all data underlying the findings in their manuscript fully available?

Reviewer #2: Yes

5. Is the manuscript presented in an intelligible fashion and written in standard English?

Reviewer #2: Yes

6. Review Comments to the Author

Reviewer #2: Considering the changes made by the authors i have no further comments. Still I mantain my reservations about the novelty of your study and the extrapolation of the results for other teams.

7. PLOS authors have the option to publish the peer review history of their article (what does this mean?). If published, this will include your full peer review and any attached files.

Reviewer #2: No

---

## [Editor Report · Acceptance letter]

29 Oct 2020

PONE-D-19-20247R2 

Potential prognostic factors for hamstring muscle injury in elite male soccer players: a prospective study 

Dear Dr. Tschan:

I'm pleased to inform you that your manuscript has been deemed suitable for publication in PLOS ONE. Congratulations! Your manuscript is now with our production department. 

Kind regards, 

on behalf of

Dr. Yumeng Li 

Academic Editor

PLOS ONE